# MID-VISION FEEDBACK

**Michael Maynord, Eadom Dessalene, Cornelia Fermüller, and Yiannis Aloimonos**
Department of Computer Science
University of Maryland, College Park
College Park, MD 20742, USA
`{maynord,edessale,fermulcm,jyaloimo}@umd.edu`

## ABSTRACT

Feedback plays a prominent role in biological vision, where perception is modulated based on agents' evolving expectations and world model. We introduce a novel mechanism which modulates perception based on high level categorical expectations: Mid-Vision Feedback (MVF). MVF associates high level contexts with linear transformations. When a context is "expected" its associated linear transformation is applied over feature vectors in a mid level of a network. The result is that mid-level network representations are biased towards conformance with high level expectations, improving overall accuracy and contextual consistency. Additionally, during training mid-level feature vectors are biased through introduction of a loss term which increases the distance between feature vectors associated with different contexts. MVF is agnostic as to the source of contextual expectations, and can serve as a mechanism for top down integration of symbolic systems with deep vision architectures. We show the superior performance of MVF to post-hoc filtering for incorporation of contextual knowledge, and show superior performance of configurations using predicted context (when no context is known a priori) over configurations with no context awareness. [1]

## 1 INTRODUCTION

In most contemporary computer vision architectures information flows in a single direction: from low-level of pixels up to high level abstract concepts (e.g., object categories) - such architectures are termed *feed-forward* architectures. In general, each successive layer of the network contains more abstract representations than the previous, and the representational hierarchy mirrors the architectural hierarchy. It is also possible to introduce *top-down connections* into the network architecture, introducing high level information into processes involving lower levels of abstraction in a process of *feedback*.

Feedback plays a primary role in biological vision; in fact, the majority of neural connections in the visual cortex are top-down, rather than bottom-up, connections (Markov et al., 2014). These top-down connections are thought to convey information of higher level *expectation*, and neurons of the visual cortex use both higher level expectation as well as lower level visual information in producing their representations. Expectations in biological systems arise from continuous engagement with the environment. In Computer Vision, this is reflected in the paradigm of Active Vision (Bajcsy, 1988; Fermüller & Aloimonos, 1995), where perception is framed as an active problem involving evolving world models.

The task of producing mid-level visual representations Teo et al. (2015a;b); Xu et al. (2012); Nishigaki et al. (2012) from low level input is under-constrained - many plausible mid-level interpretations may be consistent with input. To give an intuition for how understanding of context can impact perception of mid-level features consider Figure 1 - characteristics of shrews and kiwi differ, but may be similar enough to be confused without context. Top-down feedback - from high level context to mid-level visual features - provides a "map" for mid-level processes, constraining it towards high level consistency.

---

[1]Code will be available at: https://github.com/maynord/Mid-Vision-Feedback

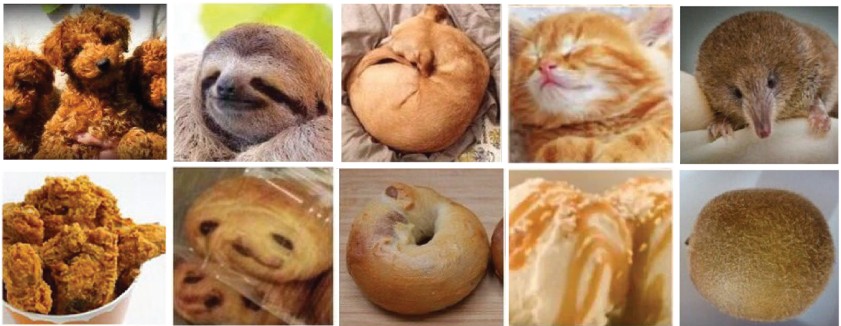

Figure 1: Here we illustrate images cropped to exclude context. At first glance, due to similarities in color, texture, and pattern, images from the top row (animate) may appear to be of the same class as those of the bottom row (inanimate). With an understanding of the difference of context, upon closer inspection, it is clear that there are meaningful lower level feature differences.

Introduction of contextual knowledge through feedback is superior to post-hoc application of contextual knowledge, e.g. through discarding interpretations (classifications, here) which are not context consistent. We demonstrate this point empirically.

Interpretations selected after post-hoc filtering for context consistency will still be built upon underconstrained mid-level features. Furthermore, in contrast to post-hoc filtering, feedback naturally allows for detection of out-of-context objects, as feedback functions through biasing of visual representations rather than filtering. It is valuable for methods to allow for out-of-context detections, even when biasing against them, as out-of-context objects on occasion appear (e.g., a tree in an office setting).

CNNs have a natural tendency towards *decoupled* representations - representations with a tendency for feature vector angle to correspond to characteristic type (e.g., "fuzzy"), and for feature vector magnitude to correspond to characteristic variation or degree (Liu et al., 2018) (e.g., "very fuzzy" / "not fuzzy") (See Figure 2 for an illustration). This opens up a couple of possibilities in terms of directly manipulating feature representations: 1) We can differentiate between axes with different associations to high level contexts, 2) we can control magnitudes of characteristics through amplifying and dampening axes associated with those characteristics. That is, w.r.t. point #1, as CNNs produce representations which are, to a degree, separated by angle, certain axes will be more associated with some higher level contexts over others. Also, w.r.t. point #2, amplifying characteristics associated with a higher level context increases the likelihood of interpreting input as conforming to that context; dampening characteristics associated with that context reduces the likelihood of interpreting input as conforming to that context.

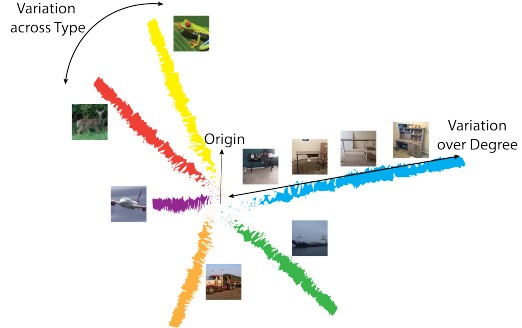

Figure 2: Artist illustration of decoupled representations, as may be learned by a CNN. For illustration we annotate characteristics with visually recognizable categories, though in this work we exploit the tendency towards decoupled representations at lower levels. Feature vector angle corresponds to characteristic type, while feature vector magnitude corresponds to within characteristic variation or degree. (Liu et al., 2018) observe that CNNs produce decoupled representations, and derive a similar illustration over MNIST by setting a convolution operator's dimension to 2.

We present a principled method to feedback - Mid-Vision Feedback (MVF), illustrated in Figure 3 - allowing the biasing of mid-level feature representations in networks such as CNNs towards conformance with high level categorical expectations. This approach is comprised of two components: 1) linear transforms (*affine transformations*), and 2) *orthogonalization bias*.

Affine transformations enable direct control over the feature vectors at the *injection level* - the level into which feedback is being inserted. If these vectors have been trained with a bias towards orthogonality w.r.t. contexts, then this allows for affine transformations to manipulate features associated with context presence or absence with less impact on other features.

The orthogonalization bias is introduced to increase the independence between contexts, so they can be manipulated with less interference. This bias is introduced at the injection level. This does not negatively impact the representational power or performance of the base network. This orthogonality bias is introduced across contexts - e.g., mid-level feature vectors associated with *animate* contexts can be biased towards orthogonality w.r.t. mid-level features associated with *inanimate* contexts. Due to the resulting greater angular separation between features associated with different contexts, this biasing allows greater control over facets of mid-level representation which are meaningful to higher level contexts.

MVF then functions as follows. During runtime a high level *context expectation* is associated with input. This expectation is used in biasing mid-level visual features through use of an affine transformation associated with the context of that expectation. This selects for characteristics associated with this context. These affine transformations are better enabled as a consequence of the disentanglement of such characteristics at the injection level, effected through introduction of the orthogonalization bias during training.

Feedback then enables a synergy between high level categorical interpretations and mid-level visual feature representations, bridging the signal-symbol gap in both directions. This approach to incorporation of context expectations is controlled. This differs from an approach of connecting the upper level fully connected layers of a network directly to lower level convolutional levels in a scheme which includes neither categorical representations nor biasing w.r.t. said categorical representations.

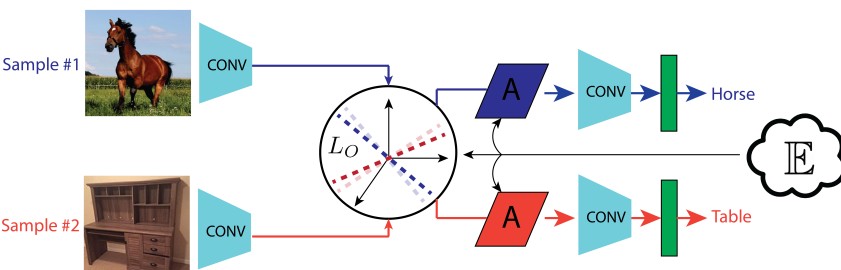

Figure 3: MVF structure. As training involves a contrastive loss ($L_O$) between samples belonging to different contexts, the base network is shown twice in the figure: on the top when fed with a horse image, and on the bottom when fed with a desk image. *Affine transformations* (linear transformations) are inserted into a conventional CNN architecture. These affine transformations modulate feature representations at injection levels (network levels over which affine transformations are applied) in accordance with high level *context expectations* ($E$). $L_O$ is inserted to put pressure on the network to help separate feature representations at the injection levels according to context, better enabling manipulation by affine transformation. During training, samples from different contexts are fed through the network in pairs. The context expectations determine the selection of appropriate affine transformations.

MVF both employs feedback from categorical knowledge and is agnostic w.r.t. the source of that categorical knowledge - i.e., it is not a requirement that context expectations be produced from the same network. As a consequence of this, MVF allows for interfacing with larger symbolic systems - e.g., models of scenes employing graphical models over scene elements and categories. This top-down synergy across the signal-symbol gap opens up a wide range of applications.

The rest of this paper is structured as follows: In Section 2 we cover related work; in Section 3 we detail methods; in Section 4 we cover experiments; and, in Section 5 we conclude.

## 2 RELATED WORK

### 2.1 BIOLOGY, FEEDBACK, AND PARALLELS TO COMPUTER VISION

Previous works ( (Markov et al., 2014), (Gilbert & Sigman, 2007), Kreiman & Serre (2020), (Gilbert & Li, 2013), and (Paneri & Gregoriou, 2017)) have explored the importance of feedback connections in biological sensory perception. Further work ((Liao & Poggio, 2016) and (van Bergen & Kriegeskorte, 2020)) draw connections between feedback in computer vision architectures and the primate visual cortex. (Tang et al., 2018) show that feedforward CNNs are not robust to occlusion, unlike in human perception, but that adding recurrence improves occlusion robustness. (Lotter et al., 2016) introduce PredNet, a network based on predictive coding, and demonstrate benefits on the task of self-supervised frame prediction.

There is good reason to believe that modeling characteristics of biological vision in computer vision architectures will benefit computer vision (e.g., (Medathati et al., 2016; Teo et al., 2015c)). For example, (Linsley et al., 2020b) demonstrate a network with top-down connections which aligns with human perception of visual illusions, where feedback aids in prioritizing object boundary contours over simple edge contours. (Linsley et al., 2020a) shows how recurrent hierarchical feedback model can improve segmentation. (Konkle & Alvarez, 2020) introduce instance-prototype contrastive learning, and show that self-supervised models can learn representations which are more brain-like than supervised models. (Li et al., 2021) introduce Contrastive Clustering, showing a benefit to instance- as well as cluster-level contrastive loss in clustering.

(Long et al., 2018) demonstrate large scale organization of the cortex based on mid-level visual features (below the level of object recognition), including those associated with *animacy* vs. *inanimacy*. (Jagadeesh & Gardner, 2022) argue that representations in category selective regions of the visual cortex encode a basis representation for texture, rather than objecthood representations. (Harrington & Deza, 2021) demonstrate that constraining networks to be robust to adversarial input produces network representations more in-line with human visual perception, and argue for the use of texture summary statistic representations.

### 2.2 FEEDBACK IN EXISTING COMPUTER VISION METHODS

The conventional approach to feedback in computer vision is the use of recurrent connections. (Caswell et al., 2016) introduce recurrent connections into shallow CNN architecture for image classification. (Pinheiro & Collobert, 2014) employ recurrency over convolutions for the purpose of enabling lateral information flow in the task of segmentation. (Zamir et al., 2016) instantiate feedback through an RNN architecture which iteratively refines prediction categories from coarse to specific.

Alternatives to conventional recurrent connections for feedback include (Hu & Ramanan, 2016), which explore convolutions with hierarchical rectified Gaussians to enable top-down as well as bottom-up information flow, and apply them to the task of keypoint localization under occlusion. Additionally, (Yao et al., 2012) apply a graphical model over scene representations, allowing higher and lower level decisions to influence each other.

## 3 METHODS

With MVF we seek a feedback mechanism which allows us to directly bias lower level feature representations based on categorical higher level context expectations. This involves top-down interaction across two levels of abstraction: 1) high level contexts $c_i \in C$, 2) mid-level features $f_i \in F$.

The structure of MVF is illustrated in Figure 3, and the loss and training formulation given in Section 3.1. Context expectations sit at a level of abstraction above the classes of network output, and are used in selecting affine transformations placed above the output of injection levels. When applied to injection level output, the affine transformations bias injection level feature representations towards conformance with the associated context expectation, as illustrated in Figure 4. Injection level output is made more amenable to manipulation according to context through introduction of a contrastive loss $L_O$, introducing a bias towards orthogonalization across context.

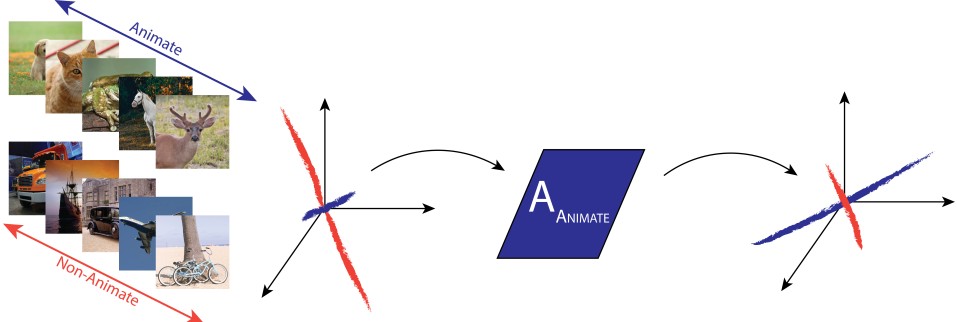

Figure 4: Illustration of the effect of affine transformation application over injection level features. Shown are images from two contexts, and the injection level feature representations for those contexts. If the dimension of the feature map of the injection level is HxWxC (Height x Width x Channel), this vector space is C-dimensional. The feature vectors are color coded according to context. After feature vector modulation through application of an affine transformation, the characteristics of the context associated with that affine transformation are more prominent. For example, with reference to Figure 1, if it is known that the context is one involving animals, rather than food, then characteristics may be biased towards fur interpretations, rather than kiwi fuzz or pastry surfaces.

Affine transformations are applied over the features of the injection level for the purpose of amplifying or dampening certain characteristics. This process aligns mid-level representations towards conformance with higher level context expectation. The affine transformations are made more effective through the disentanglement of characteristics at the injection level produced by the orthogonalizing bias.

During test time the CNN runs as a single stream (without the connection across the streams of multiple samples which the contrastive loss introduces), and a context expectation selects the affine transformation to apply over the feature vectors of the injection level. This expectation can come from any source - in Section 4 we compare performance across network produced context expectations and ground truth context expectations.

### 3.1 Loss and Training

Training is broken into two stages, as detailed in Table 1. In the first, the base network is trained on its own, and features are biased towards orthogonality at the injection levels. In the second stage, the learning rate for the network parameters is reduced, and affine transformations are added to the injection levels according to the context categories of input samples, initialized to identity matrices with added random noise, and given their own optimizers and learning rate. In the first stage, gradients backpropogate through the base network only, bypassing the affine transformations; in the second stage gradients pass through both the base network and the affine transformations. In each stage we employ batches containing equal proportions of samples belonging to each context.

Training is broken into two stages for a few reasons: 1) this division allows the possibility of using pretrained networks and training affine transformations in injection levels with no modification to the base network ($\lambda = 0$ and $\eta_{N_2} = 0$), 2) allows fine-tuning of pretrained networks ($\lambda > 0$ and/or $\eta_{N_2} > 0$), 3) we find that starting training of affine transformations after the feature representations have had a chance to converge helps the affine transformations train - intuitively, the affine transformations have to adapt to less of a moving target. We allow the network parameters to continue to train after affine transformations have been introduced - at a reduced learning rate - as we find that this benefits performance.

To illustrate this process, consider a batch containing a *horse* and a *desk* image, *horse* belongs to animate while *desk* belongs to inanimate context. Both the images are fed in the same batch to the network during training. When training is in Stage 1, each image passes through the base network, sans affine transformations. The feature representations of the injection levels for horse and desk are then connected to each other via $L_O$, and biased towards orthogonality with respect to each other. When training progresses to Stage 2, the $L_O$ contrastive connection is removed,

and affine transformations are inserted according to context expectation. Both network and affine transformation parameters are then updated according to gradients that pass through both affine transformations and network parameters.

We find that the angles between mid-level features associated with higher level contexts can be increased to a much greater degree than they would be without this orthogonalizing loss, as measured by cosine similarity, without appreciable negative impacts on performance. See Figure 6 in the Appendix for an illustration of the extent to which this cosine loss is reduced when introducing this orthogonalizing loss.

| | Stage 1 | Stage 2 |
|---|---|---|
| Loss | $L_1(Y, P, F) = \lambda * L_O(F, Y) + CE(Y, P)$ | $L_2(Y, P, F) = CE(Y, P)$ |
| Optimizers | $O_N(\eta_{N_1}, L_1)$ | $O_N(\eta_{N_2}, L_2) + \sum_{i=0}^{|C|} O_{A_{c_i}}(\eta_C, L_2)$ |

Table 1: Definition of losses ($\{L_1, L_2\}$), and their use in optimizers across Stages 1 and 2 of training. $Y, P$, and $F$ correspond to sample labels, predictions, and injection level features, respectively. $C$ is the context set. $L_O(F, Y)$ is the orthogonalization bias loss described in Equation 1, $\lambda$ (intermediate loss scaling) is a scaling term, and $CE(Y, P)$ is cross entropy loss. Optimizer $O_N$ optimizes over loss $L_j$ according to learning rate $\eta_{N_j}$ for training stage $j$. Optimizer $O_{A_{c_i}}$ is an optimizer applied over only the parameters of the affine transformation associated with context $c_i$, for network learning rates $\eta_{N_1}, \eta_{N_2}$, affine transformation learning rate $\eta_C$, and losses $L_1, L_2$.

$$L_O(F, Y) = \frac{1}{|S_{F,Y}|} \sum_{(f_1, f_2) \in S_{F,Y}} max(0, \frac{f_1 \cdot f_2}{\|f_1\|\|f_2\|}) \tag{1}$$

Where

$$S_{F,Y} = \{(f_1, f_2)|f_i \sim U(F_{c_i}), Y_C(f_1) \neq Y_C(f_2), I(f_1) = I(f_2)\} \tag{2}$$

Where

$$F_{c_i} = \{f|Y_C(f) = c_i\} \tag{3}$$

Where $|S_{F,Y}|$ is a method hyper-parameter, $Y_C(f)$ is the context of the sample from which feature vector $f$ was produced, $I(f)$ is the injection level from which $f$ was taken, and $U(A)$ is the uniform probability distribution over elements of $A$.

With $L_O$ we wish to separate the angles of features associated with different contexts, in order to better enable manipulation through affine transformations. This can be seen as an exacerbation of CNN's natural tendency towards decoupled representations - where feature type has a tendency to group according to feature vector angle - through a structuring of characteristics' feature vector angles according to higher level context.

We do this through a cosine loss ($\frac{A \cdot B}{\|A\|\|B\|}$) applied to the features of the injection level. As we wish to control the network representation in terms of context expectation, we apply this loss across context. Figure 7 in the Appendix illustrates the behavior produced by the orthogonalizing bias.

## 4 EXPERIMENTS

Here we cover experiments demonstrating the utility of our feedback method. We perform evaluations over CIFAR100 (Krizhevsky et al., 2009), ImageNet (Deng et al., 2009), and the Caltech UCSD Birds Data set (Birds) (Wah et al., 2011), all with multiple context splits - these datasets are

| | CIFAR | | | | | ImageNet | | | Birds | | |
|---|---|---|---|---|---|---|---|---|---|---|---|
| | 1 | 2 | 3 | 4 | Full | 1 | 2 | 3 | 1 | 2 | 3 |
| VGG | 87.5 | 75.5 | 87.1 | 88.7 | 72.64 | 88.51 | 86.5 | 88.2 | 70.37 | 71.02 | 71.80 |
| VGG (PF) | **89.1** | **76.8** | **88.9** | **89** | **74.32** | **90** | **88.17** | **89.1** | **71.25** | **76.6** | **74.7** |
| ViT | 81.77 | 70.8 | 75.32 | 81.1 | 71.1 | 73.66 | 76.03 | 79.92 | 70.25 | **69.5** | 71 |
| ViT (PF) | **83.17** | **76.97** | **78.0** | **81.9** | **74.29** | **74.0** | **79.75** | **80.24** | **70.3** | 69.1 | **71.2** |
| CNN | 71.6 | 57.1 | 70.34 | 71.3 | 52 | 57.8 | 70.23 | 64.8 | x | x | x |
| CNN (PF) | **74.1** | **59.5** | **71.91** | **74** | **59.8** | **58.06** | **70.6** | **66** | x | x | x |

Table 2: Object classification accuracies (shown in percentages) across model and dataset splits. Comparison between base model and feedback based on Predicted Feedback (PF) context, where the context fed down the network is first predicted by the same network (accuracies for context prediction are given in Table 6 in the Appendix). The simple CNN was not applied over Birds data as its expected input resolution is small.

| | CIFAR | | | | | ImageNet | | | Birds | | |
|---|---|---|---|---|---|---|---|---|---|---|---|
| | 1 | 2 | 3 | 4 | Full | 1 | 2 | 3 | 1 | 2 | 3 |
| VGG | 87.5 | 75.5 | 87.1 | 88.7 | 72.64 | 88.51 | 86.5 | 88.2 | 70.37 | 71.02 | 71.80 |
| VGG (GTM) | 92.8 | 81.2 | **89.8** | 93.4 | **74.2** | 89.6 | **89.4** | 90.44 | 75.32 | 77.26 | 74.82 |
| VGG (GTF) | **93.4** | **81.4** | 88.8 | **93.8** | 74 | **89.8** | 89.07 | **91.36** | **78.77** | **79.23** | **76.17** |
| ViT | 81.77 | 70.8 | 75.32 | 81.1 | 71.1 | 73.66 | 76.03 | 79.92 | 70.25 | 69.5 | 71 |
| ViT (GTM) | 89.8 | 78.1 | **79.3** | 87.8 | 73 | 76.5 | **79.81** | **83.91** | - | - | - |
| ViT (GTF) | **90.22** | **78.4** | 79.2 | **89.9** | **73.1** | **76.6** | 77.66 | 80 | - | - | - |
| CNN | 71.6 | 57.1 | 70.34 | 74 | 52 | 57.8 | 70.23 | 64.8 | x | x | x |
| CNN (GTM) | 81.3 | 64.4 | 71.37 | 79.5 | 55.7 | 61.95 | 71.45 | **72.3** | x | x | x |
| CNN (GTF) | **83.2** | **64.4** | **71.9** | **80.7** | **62.1** | **62.85** | **73** | 71.2 | x | x | x |

Table 3: Object classification accuracies across model and dataset splits. Comparison between base model, Masking based on Ground Truth context (GTM), and Feedback based on Ground Truth context (GTF). The simple CNN was not applied over Birds data as its expected input resolution is small.

described in Section B of the Appendix and derived based on the CIFAR-100 superclasses, and the attribute labels provided in the CUB dataset. We base splits on this information in order to evaluate over standard divisions in the data. We evaluate our method using both ground truth context expectations (**GT Feedback**), as well as context expectations derived from a network of the same structure as the base network (**Pred Feedback**). All experiments are conducted using a 6-layer CNN base architecture, a VGG-16 network (Simonyan & Zisserman, 2014), and a Transformer model (Tu et al., 2022), with variants including added affine transformations for feedback runs. The hyperparameters to the modifications made to each of the base architectures for feedback incorporation are described briefly in Section H in the Appendix, and described in detail in Section I in the Appendix. We show confusion matrices for a 10-class context split over CIFAR100 for both ground truth context expectations and network predicted context expectations, in Figures 5a and 5b, respectively. In Sections 4.1 and 4.2 evaluation involves comparisons to base architectures tuned to maximize base architecture performance, and Section 4.3 presents an evaluation where both base architecture and feedback implementation are tuned to maximize the benefit of feedback, giving insight into extent of potential benefit to feedback.

## 4.1 ORACLE EVALUATION

The ground truth context model (**GT Feedback**) assumes access to the ground truth context belonging to each input sample during training and test time, using contextual knowledge to index into the affine transformations for application over mid-level features as described in Section 3. Here we evaluate the extent to which our complete framework outperforms 1) a base architecture mirroring that of our framework, and 2) the same base architecture where a hard masking operation using the ground truth contextual knowledge is applied over its class-level outputs. The **GT Masking** baseline corresponds to the same base architecture where a hard mask of $c_g \in \{0, 1\}^k$ is applied over the output of the network, where $k$ corresponds to the number of classes and $c_g$ takes on a value of 1 for

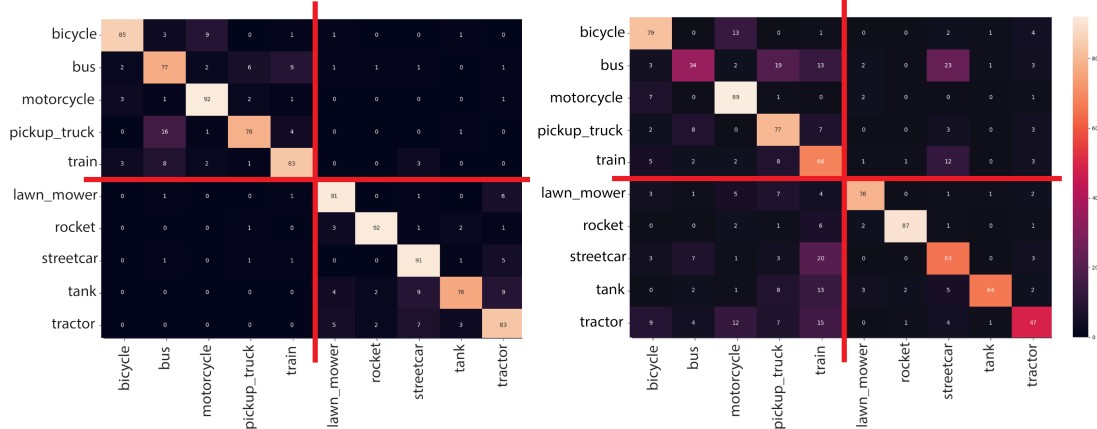

(a) With orthogonalization bias and affine transforma-
tions.

(b) With neither orthogonalization bias nor affine trans-
formations.

Figure 5: Confusion matrices for a our simple architecture (see Appendix ) with and without feedback (orthogonalization bias and with application of affine transformations) - subfigure a and b respectively - over the vehicles 1 vs vehicles 2 split of CIFAR100. The first 5 rows correspond to the first context, and the next 5 rows correspond to the second context. Note that cross context confusion (quadrants 1 and 3 of the confusion matrices) is significantly reduced with feedback.

within-context classes and 0 for out-of-context classes. Results for ground truth context evaluation are shown in Figure 3.

## 4.2 PREDICTED CONTEXT EVALUATION

Table 2 shows performance of base architectures with and without feedback, when context is predicted rather than provided. Context prediction is performed through the addition of a second logits classification head to the base networks. This head is trained in conjunction with the object classification head using ground truth context labels. Performance for context prediction is shown in Table 6 in the Appendix.

## 4.3 MAXIMIZING FEEDBACK GAIN

We here present an evaluation where we tune for maximizing margins of feedback performance over base model performance, tuning both base model and feedback parameters. This evaluation provides insight into the degree to which feedback is capable of improving performance. We tune over the following hyperparameters to maximize the margins between feedback models relying on ground truth and the base models without feedback: first stage learning rate, weight decay, second stage learning rate, and affine learning rate. See the Appendix for precise values.

## 4.4 DISCUSSION

We evaluate the utility of feedback under two scenarios: 1) where context is known (Table 3), 2) where context is not known (Table 2). This provides insight both on the performance of the feedback mechanism in ideal cases (providing an upper bound on the utility of feedback), as well as realistic cases with imperfect information derived from the same network. We observe consistently positive results in each case. We also compare feedback to a strong alternative of masking out (excluding) predictions not associated with the ground truth context. Consistent with Figure 5 superior performance of feedback over masking demonstrates that it improves modeling beyond simply removing context inconsistent predictions. Results were presented both in comparisons where base architectures were tuned to maximize accuracy, and comparisons where base and feedback models were tuned to maximize the benefit of feedback. Margins are much higher when tuning to maximize

| | CIFAR | | | | | ImageNet | | | Birds | | |
|---|---|---|---|---|---|---|---|---|---|---|---|
| | 1 | 2 | 3 | 4 | Full | 1 | 2 | 3 | 1 | 2 | 3 |
| VGG | 87.1 | 70.4 | 83.8 | 85.9 | 70.5 | 85.3 | 85.0 | 85.4 | 70.0 | 72.8 | 68.4 |
| VGG (PF) | **89.9** | **77.4** | **85.8** | **88.2** | **77.0** | **88.9** | **86.1** | **89.5** | **75.6** | **77.4** | **77.8** |
| ViT | 82.1 | 56.8 | 78.7 | 82.4 | 65.4 | 74.0 | 77.0 | 78.6 | 66.6 | 54.2 | 52.7 |
| ViT (PF) | **86.9** | **59.0** | **82.6** | **85.7** | **68.0** | **76.0** | **79.2** | **83.9** | **69.9** | **65.1** | **64.3** |
| CNN | 67.5 | 55.9 | 66.1 | 63.2 | 49.9 | 53.6 | 55.9 | 58.4 | x | x | x |
| CNN (PF) | **77.1** | **60.3** | **70.6** | **71.3** | **52.9** | **58.0** | **61.9** | **63.9** | x | x | x |

Table 4: Results where we **Maximize Feedback Gain** (see Subsection 4.3). Object classification accuracies across model and dataset splits, where comparisons are between base models and feedback based on Predicted Feedback (PF) context, where the context fed down the network is first predicted by the same network. The simple CNN was not applied over Birds data as its expected input resolution is small.

| | CIFAR | | | | | ImageNet | | | Birds | | |
|---|---|---|---|---|---|---|---|---|---|---|---|
| | 1 | 2 | 3 | 4 | Full | 1 | 2 | 3 | 1 | 2 | 3 |
| VGG | 87.1 | 70.4 | 83.8 | 85.9 | 70.5 | 85.3 | 85.0 | 85.4 | 70.0 | 72.8 | 68.4 |
| VGG (GTM) | 91.7 | 75.4 | 88.2 | 90.0 | 72.2 | 86.6 | 87.3 | 88.6 | 76.9 | 77.3 | 74.6 |
| VGG (GTF) | **93.3** | **81.5** | **88.4** | **91.9** | **76.4** | **88.8** | **87.5** | **91.1** | **80.2** | **79.1** | **80.4** |
| ViT | 82.1 | 56.8 | 78.7 | 82.4 | 65.4 | 74.0 | 77.0 | 78.6 | 66.6 | 54.2 | 52.7 |
| ViT (GTM) | 87.2 | 64.0 | 82.5 | 86.3 | 68.1 | 77.5 | 80.0 | 82.2 | 70.0 | 61.8 | 61.8 |
| ViT (GTF) | **91.1** | **66.2** | **85.7** | **90.5** | **71.9** | **79.5** | **81.7** | **86.0** | **73.9** | **65.1** | **65.6** |
| CNN | 67.5 | 55.9 | 66.1 | 63.2 | 49.9 | 53.6 | 55.9 | 58.4 | x | x | x |
| CNN (GTM) | 79.3 | 64.8 | 73.5 | 74.5 | 51.4 | 59.6 | 62.9 | 66.6 | x | x | x |
| CNN (GTF) | **82.4** | **69.0** | **75.6** | **76.9** | **56.0** | **62.2** | **64.7** | **69.8** | x | x | x |

Table 5: Results where we **Maximize Feedback Gain** (see Subsection 4.3). Object classification accuracies across model and dataset splits, where comparisons are between base models, Masking based on Ground Truth context (GTM) and Feedback based on Ground Truth context (GTF). The simple CNN was not applied over Birds data as its expected input resolution is small.

the benefit of feedback, and give insight into the extent to which feedback is capable of providing benefits in the best case.

## 5 CONCLUSION

We have presented an argument for the utility of feedback in vision. Feedback is 1) prominent in biological vision, with the majority of neural connections in the cortex consisting of feedback connections, 2) allows better constraining of under-constrained processes of abstraction, 3) allows for the online adaptation of vision systems towards alignment with high level understanding of the world. We leverage the fact that CNNs have a tendency towards decoupled representations, exacerbating the separation of mid-level features associated with different higher level contexts. This allows better direct manipulation of the level at which feedback is introduced, minimizing collateral effects on characteristics not being selected for. In contrast to post-hoc filtering of interpretations for consistency with context expectations, MVF allows for cross-context detections and produces higher accuracies. MVF involves a top-down bridging of the signal-symbol gap, making it applicable to a range of applications. In the future this work will be extended to localization, e.g. object detection or semantic segmentation, as well as used in embodied contexts Fermüller & Maynord (2022) with an active agent.

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

| | CIFAR | | | | | ImageNet | | | Birds | | |
|---|---|---|---|---|---|---|---|---|---|---|---|
| | 1 | 2 | 3 | 4 | Full | 1 | 2 | 3 | 1 | 2 | 3 |
| VGG | 91.65 | 95.34 | 94.62 | 95.11 | 96.43 | 98 | 97.9 | 97.8 | 81.19 | 90.3 | 75.3 |
| ViT | 90.32 | 96.17 | 93.46 | 94.17 | 96.2 | 98.5 | 94.3 | 97 | 80.4 | 87.19 | 80.29 |
| CNN | 82.9 | 85.5 | 88.24 | 85.5 | 87.25 | 89.33 | 90 | 92.46 | x | x | x |

Table 6: Context classification accuracies (shown as percentages, and used in Section 4.2) across model and dataset splits. The simple CNN was not applied over Birds data as its expected input resolution is small.

## A  APPENDIX

### ABSTRACT

We here present Appendix. Section B describes datasets; Section D presents the base CNN architecture used in experiments; Section I details tuned parameters; and, Section K presents data splits.

## B  DATASETS

### B.1  CIFAR100

We adopt CIFAR100 for the 1) high-levels of visual ambiguity due to low resolution, 2) existence of several distinct "superclasses" consisting of a roughly equal number of classes, and 3) the cross-context confusion across classes highly similar in appearance (e.g., sharks and dolphins). We adopt the official training and test split for the CIFAR100 dataset. Each class contains exactly 500 training images and 100 testing images, with each superclass consisting of 5000 training images and 1000 testing images. We use the CIFAR100 superclasses in constructing context splits. Split 1: Vehicles 1 vs. Vehicles 2, Split 2: Household Devices vs. Furniture, Split 3: Aquatic Mammals vs. Fish. For full class breakdown see Appendix Section K.

### B.2  IMAGENET

We adopt ImageNet for several of the aforementioned reasons above, as well as its generality in that it spans 1000 unique classes. Each class contains variable number of images - we designate $80\%$ of the images in each class for training, but only $2\%$ for testing due to the computational cost incurred by the high number of images and the need for frequent testing.

We employ the following context split over ImageNet, designed to be similar to CIFAR100 splits, the full class breakdown of which is given in Appendix: Split 1: Household Devices vs. Furniture, Split 2: Aquatic Mammals vs. Fish, Split 3: Vehicles 1 vs. Vehicles 2.

## C  BASE CNN

Figure 8 illustrates the base CNN model (apart from VGG and ViT), for which performance is reported in Tables in the main paper.

## D  CONTEXT PREDICTION

See Table 6 for accuracy on context prediction used in Section 4.2 and Table 2; see Table 7 for accuracy on context prediction used in Section 4.2 and Table 2.

## E  IMAGENET-C EVALUATION

Table 8 provides accuracy of testing on ImageNet-C, with models trained for **Maximizing Feedback Gain** over standard ImageNet. Accuracy trends are consistent with trends presented in Section 4.3.

| | CIFAR | | | | | ImageNet | | | Birds | | |
|---|---|---|---|---|---|---|---|---|---|---|---|
| | 1 | 2 | 3 | 4 | Full | 1 | 2 | 3 | 1 | 2 | 3 |
| VGG | 93.7 | 91.9 | 89.6 | 90.4 | 96.1 | 97.8 | 95.9 | 97.6 | 79.0 | 89.4 | 80.3 |
| ViT | 90.32 | 96.17 | 93.46 | 94.17 | 96.2 | 98.5 | 94.3 | 97 | 80.4 | 87.19 | 80.29 |
| CNN | 82.3 | 86.1 | 88.8 | 81.0 | 90.2 | 85.6 | 87.6 | 90.6 | x | x | x |

Table 7: Context classification results of the runs **Maximizing Feedback Gain** (see Subsection 4.3 across model and dataset splits. The simple CNN was not applied over Birds data as its expected input resolution is small.

| | ImageNet | | |
|---|---|---|---|
| | 1 | 2 | 3 |
| CNN | 39.5 | 41.8 | 42.8 |
| CNN PF | 42.3 | 47.6 | 48.0 |
| CNN GTM | 45.5 | 48.7 | 52.4 |
| CNN GTF | 47.7 | 50.1 | 53.9 |
| ViT | 58.3 | 61.7 | 64.1 |
| ViT PF | 61.3 | 64.8 | 69.5 |
| ViT GTM | 62.1 | 64.6 | 66.5 |
| ViT GTF | 64.4 | 66.4 | 68.5 |
| VGG | 69.7 | 69.3 | 69.8 |
| VGG PF | 74.5 | 71.8 | 75.3 |
| VGG GTM | 70.9 | 72.3 | 73.2 |
| VGG GTF | 73.3 | 71.9 | 75.2 |

Table 8: Object classification accuracies across ImageNet-C, where the training set is unmodified and various corruptions are applied over the test set. Comparison between base model, Masking based on Ground Truth context (GTM), feedback based on Predicted Feedback (PF) context, and Feedback based on Ground Truth context (GTF).

ImageNet-C consists of 75 common corruptions applied over ImageNet images with the intent of degrading classifier performance. We observe that drops in accuracy with respect to the original ImageNet dataset range between values of $14\%$ and $18\%$. However, performance margins between feedback and base models are overall maintained when testing over ImageNet-C.

# F    DECOUPLED REPRESENTATIONS

CNNs have a natural tendency towards *decoupled* representations. These are representations where characteristics have a tendency to be represented in such a way that feature vector angle corresponds to characteristic type, while feature vector magnitude corresponds to characteristic variation or degree (Liu et al., 2018).

# G    ORTHOGONALIZING LOSS

Figure 7 illustrates feature vector projections of the injection level under different degrees of orthogonalizing loss.

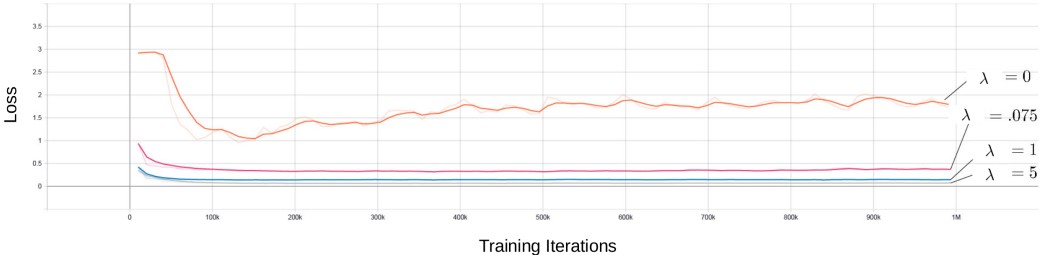

Figure 6: Shown above is a convergence plot of orthogonality loss $L_O$ over several $\lambda$ (intermediate loss scaling) values of $\{0, 0.075, 1, 5\}$.

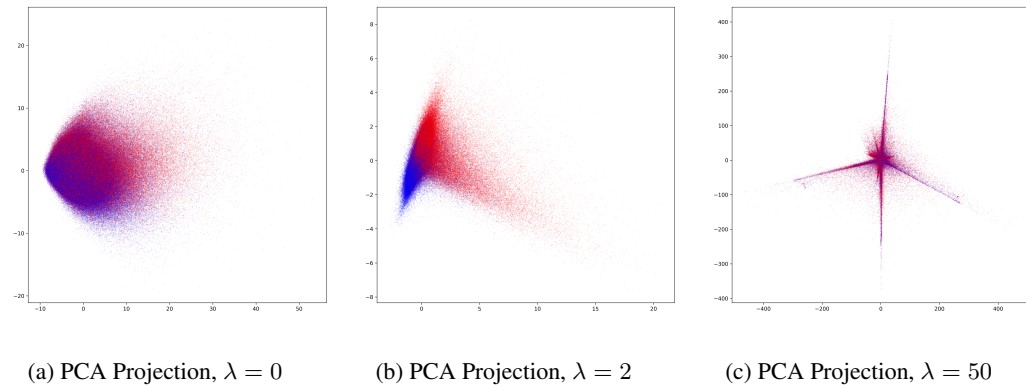

(a) PCA Projection, $\lambda = 0$     (b) PCA Projection, $\lambda = 2$     (c) PCA Projection, $\lambda = 50$

Figure 7: Illustrations of the effect of different levels of $\lambda$, the scaling term applied to orthogonalization bias $L_O$, applied to an injection level of the second to last conv op of the architecture given in the Appendix. Feature vectors of size 64 for the (animate) / (inanimate) split (see Appendix) of CIFAR100 are projected into 2 dimensions through Principal Component Analysis (PCA). Blue points correspond to features produced from animate images, and red points correspond to features produced by inanimate classes. Subfigure a illustrates context distribution when no orthogonalizing bias is used ($\lambda = 0$) - there is some degree of context separation, but it is not significant. A moderate degree of orthogonalizing bias ($\lambda = 2$) in subfigure 2 separates animate from inanimate features in the injection level. A significant degree of orthogonalizing bias ($\lambda = 50$) results in poor separation and degraded performance. See Appendix for more details, including UMAP projections.

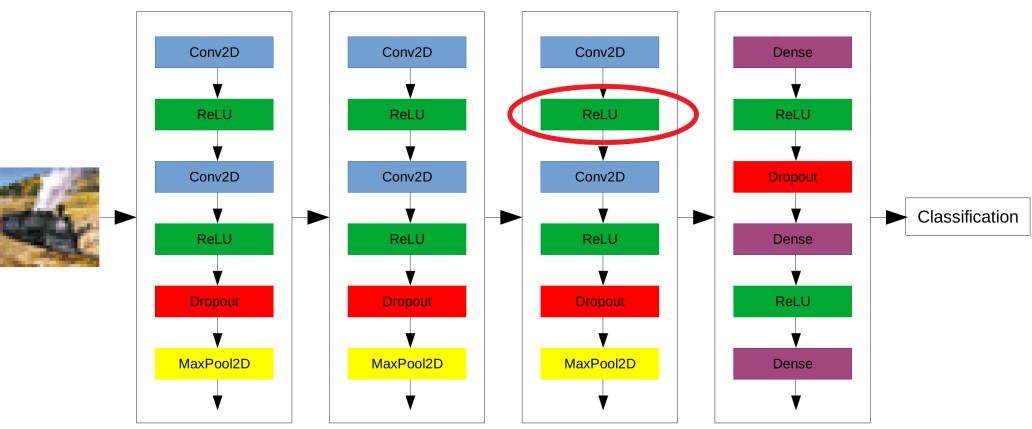

Figure 8: Illustration of the base architecture (SimpleCNN) used in these experiments. The injection level (circled in red) used in these experiments is after the ReLU of the second to last Conv2D operator.

## H    MODELS

Here we describe the architectures in which we incorporate feedback. Each model consumes 1 GPU during train and test time. For all feedback experiments we choose a $\lambda$ (intermediate loss scaling) of 1.0 (otherwise set to 0.0).

**Shallow CNN:** This model comprises a 6-layer CNN architecture, shown in Appendix, consisting of 3 by 3 shaped kernels, max-pooling applied over every other layer, and dropout ($p = 0.375$, $p = 0.1$) applied over the penultimate fully connected layer and after each convolution operation, respectively. The affine transformation is applied after the second to last convolution operation, though we observe high performance inserting the affine transformations anywhere throughout the second half of the architecture. We train the first stage for roughly 5 million iterations for all splits. A learning rate of 0.001 is chosen for the training of the base network during the first stage, and a learning rate of $5 * 10^{-5}$ is chosen for the learning rate of the base network during the second stage, whereas the affine transformation learning rate is set to $1 * 10^{-3}$.

For the **Maximizing Feedback Gain** hyperparameters, we adopt a learning rate of $2e - 4$, a weight decay of 0.0, a second stage learning rate of $1e - 6$, and an affine learning rate of 0.005.

**VGG:** Here we adopt a VGG-16 network with pre-trained weights over ImageNet. The VGG network consists of 16 layers consisting of convolution and max-pooling operations. The affine transformation is applied after the eleventh convolution operation, though we observe high performance inserting the affine transformations anywhere throughout the last six layers. We train the first stage for roughly 1.5 million iterations for all splits, until smooth convergence. A learning rate of $5 * 10^{-6}$ is chosen for the training of the base network during the first stage, and a learning rate of $2.5 * 10^{-6}$ is chosen for the learning rate of the base network during the second stage, whereas the affine transformation learning rate is set to $2.5 * 10^{-3}$.

For the **Maximizing Feedback Gain** hyperparameters, we adopt a learning rate of $5e - 5$, a weight decay of 0.00075, a second stage learning rate of $5e - 6$, and an affine learning rate of 0.0005.

**Visual Transformer:** Here we adopt a variant of the Visual Transformer models (Tu et al., 2022), a general-purpose vision transformer that outperforms many related visual transformer architectures while being easy to train. The affine transformation is applied immediately after the third to last attention block. We train the first stage for roughly 2.0 million iterations for all splits, until smooth convergence. A learning rate of $1 * 10^{-3}$ is chosen for the training of the base network during the first stage, and a learning rate of $1.0 * 10^{-6}$ is chosen for the learning rate of the base network during the second stage, whereas the affine transformation learning rate is set to $1 * 10^{-3}$.

For the **Maximizing Feedback Gain** hyperparameters, we adopt a learning rate of $2e - 4$, a weight decay of 0.0, a second stage learning rate of $1e - 5$ and an affine learning rate of 0.0001.

## I    PARAMETERS

We here list parameters' tuned values not introduced in the main paper:

1. Image size: $32 \times 32$ for CIFAR100, $224 \times 224$ for ImageNet.
2. Model input image size: $32 \times 32$ for 6-layer CNN, $224 \times 224$ for VGG16. Images resized using bilinear interpolation.
3. Size of feature set selected for orthogonalization: 25.
4. Batch size: 256 (CIFAR100 splits), 64 (ImageNet splits).
5. Data augmentations: Random rotations (15 degrees), random resized crops, Random horizontal flips.
6. Feedback Base Model: ADAM's optimizer, weight decay of $7.5 \times 10^{-4}$ for both stages and both models.
7. Affine Transformation Optimizer: ADAM's optimizer, affine transformation learning rate of 0.001 for second stage training of both models.
8. Context Model: ResNet18 model with pretrained weights over ImageNet and learning rate of 0.001 using SGD optimizer.

| | CIFAR | | | | | ImageNet | | | Birds | | |
|---|---|---|---|---|---|---|---|---|---|---|---|
| | 1 | 2 | 3 | 4 | Full | 1 | 2 | 3 | 1 | 2 | 3 |
| VGG | 87.1 | 70.4 | 83.8 | 85.9 | 70.5 | 85.3 | 85.0 | 85.4 | 70.0 | 72.8 | 68.4 |
| VGG (*) | 87.4 | 70.8 | 84.6 | 87.3 | 71.7 | 86.8 | 85.8 | 86.0 | 71.2 | 73.9 | 70.0 |
| VGG (**) | 87.7 | 70.4 | 86.0 | 87.5 | 72.0 | 88.0 | 85.4 | 86.7 | 71.9 | 74.2 | 71.3 |
| VGG (Pred Feedback) | 89.9 | 77.4 | 85.8 | 88.2 | 77.0 | 88.9 | 86.1 | 89.5 | 75.6 | 77.4 | 77.8 |
| VGG (GT Feedback) | 93.3 | 81.5 | 88.4 | 91.9 | 76.4 | 88.8 | 87.5 | 91.1 | 80.2 | 79.1 | 80.4 |
| ViT | 82.1 | 56.8 | 78.7 | 82.4 | 65.4 | 74.0 | 77.0 | 78.6 | 66.6 | 54.2 | 52.7 |
| ViT (*) | 84.7 | 56.9 | 79.8 | 83.1 | 65.8 | 74.6 | 75.3 | 79.5 | 67.0 | 56.2 | 56.7 |
| ViT (**) | 85.2 | 57.1 | 80.0 | 83.0 | 67.0 | 75.9 | 75.7 | 80.0 | 67.8 | 56.7 | 57.0 |
| ViT (Pred Feedback) | 86.9 | 59.0 | 82.6 | 85.7 | 68.0 | 76.0 | 79.2 | 83.9 | 69.9 | 65.1 | 64.3 |
| ViT (GT Feedback) | 91.1 | 66.2 | 85.7 | 90.5 | 71.9 | 79.5 | 81.7 | 86.0 | 73.9 | 65.1 | 65.6 |
| CNN | 67.5 | 55.9 | 66.1 | 63.2 | 49.9 | 53.6 | 55.9 | 58.4 | x | x | x |
| CNN (*) | 71.3 | 55.9 | 67.4 | 65.3 | 50.3 | 55.3 | 55.2 | 58.5 | x | x | x |
| CNN (**) | 72.9 | 57.0 | 67.8 | 67.1 | 50.5 | 56.2 | 55.9 | 59.3 | x | x | x |
| CNN (Pred Feedback) | 77.1 | 60.3 | 70.6 | 71.3 | 52.9 | 58.0 | 61.9 | 63.9 | x | x | x |
| CNN (GT Feedback) | 82.4 | 69.0 | 75.6 | 76.9 | 56.0 | 62.2 | 64.7 | 69.8 | x | x | x |

Table 9: Object classification accuracies across models (**Maximizing Feedback Gain**, Section 4.3) and dataset splits. We additionally report numbers when the context training signal is included in training (**) vs. when the context signal is excluded from training (*), when the feedback mechanism is not incorporated (neither the affine selection nor the orthogonalization loss). This isolates the impact of the inclusion of the context label during training while removing the impact of the feedback mechanism. The difference in performance between (*) and (**) runs is attributable to the inclusion of the context signal during training. For both runs we use an affine learning rate of $0$ and a single affine transformation as opposed to $N$ affine transformations (where $N$ is the number of context divisions; we use a single affine to avoid introducing a context signal, even when affines are not trained). A context scaling loss value of $0.0$ is used for (*) and a value of $1.0$ for (**). The runs of (**), where the context signal is included but feedback is excluded, underperform w.r.t. runs where feedback is included.

## J  Context Label, Affine Learning Rate, Orthogonalizing Loss Ablation

In Table 9, we evaluate the effect on performance due simply to the introduction of the affine transformation (and the random noise introduced by its introduction), but not due to the context training labels. We report numbers from experiments where the affine operations are included in the network but: affine transformations are not trained, the context prediction head is not trained, and orthogonalizing loss is not employed. These runs are compared against identical runs where the affine transformation is not included. We observe that runs with affine transformations outperform the results of the base models (where no affine transformations are included), for two main possible reasons: 1) The drop in learning rate during the second stage of training allows accuracy to continue converging after possible plateauing, and 2) The introduction of a randomly initialized affine during the second stage introduces stochasticity potentially useful during training. This increase in performance is small in comparison to the increase due to incorporation of feedback.

## K  Data Splits

We derive context splits based on the superclass structure provided with CIFAR-100 (over both CIFAR-100 and ImageNet), and the attribute ontology provided with the CUB dataset. We base splits on this information in order to evaluate over standard divisions in the data.

### K.1  CUB-200-2011

We adopt the Caltech-UCSD-Birds dataset for several of the aformentioned reasons above, in particular for the high cross-context confusion across different species of birds highly similar in appearance. It consists of 11,788 images with 200 classes corresponding to bird species. Like the Imagenet dataset, we designate $80\%$ of the dataset for training and $20\%$ for testing.

We employ the following 3 splits over the CUB dataset, grouping images into contexts based on the listed attributes provided with the CUB dataset:

1. Migration behavior (1, 2, 3)

2. Trophic level (Carnivore, Herbivore, Omnivore)

3. Primary lifestyle (Aerial, Aquatic, Generalist, Insessorial, Terrestrial)

## K.2   SPLITCIFAR

### CIFAR100 Dataset Sub-Splits

| Split | Group | Classes |
|:---:|:---:|:---:|
| 1 | Vehicles 1
Vehicles 2 | **Bicycle, Bus, Motorcycle, Pickup truck**
**Lawn mower, Rocket, Streetcar, Tank, Tractor** |
| 2 | Household Devices
Furniture | **Clock, Keyboard, Lamp, Telephone, television**
**Bed, Chair, Couch, Table, Wardrobe** |
| 3 | Aquatic mammals
Fish | **Beaver, Dolphin, Otter, Seal, Whale**
**aquarium fish, flatfish, ray, shark, trout** |
| 4 | Small animals
Large animals | **fox, porcupine, possum, raccoon, skunk**
**bear, leopard, lion, tiger, wolf** |

Full CIFAR100 Split:

**animate** = beaver, dolphin, otter, seal, whale, aquarium_fish, flatfish, ray, shark, trout, bear, leopard, lion, tiger, wolf, camel, cattle, chimpanzee, elephant, kangaroo, fox, porcupine, possum, raccoon, skunk, baby, boy, girl, man, woman, crocodile, dinosaur, lizard, snake, turtle, hamster, mouse, rabbit, shrew, squirrel, bee, beetle, butterfly, caterpillar, cockroach, crab, lobster, snail, spider, worm

**inanimate** = orchid, poppy, rose, sunflower, tulip, bottle, bowl, can, cup, plate, apple, mushroom, orange, pear, sweet_pepper, clock, keyboard, lamp, telephone, television, bed, chair, couch, table, wardrobe, bridge, castle, house, road, skyscraper, cloud, forest, mountain, plain, sea, maple_tree, oak_tree, palm_tree, pine_tree, willow_tree, bicycle, bus, motorcycle, pickup_truck, train, lawn_mower, rocket, streetcar, tank, tractor

### K.3  SPLITIMAGENET

### ImageNet Dataset Splits

| Split | Group | Classes |
|-------|-------|---------|
| 1 | Household Devices | **analog clock, digital clock, wall clock, computer keyboard, dial telephone, table lamp, television, cellular telephone** |
|  | Furniture | **studio couch, dining table, wardrobe, folding chair** |
| 2 | Aquatic mammals | **Beaver, Dolphin, Otter, Seal, Whale** |
|  | Fish | **barracouta, eel, coho, rock beauty, anemone fish, sturgeon, gar, puffer, lionfish** |
| 3 | Devices 1 | **mountain bike, bicycle-built-for-two, school bus, moped, tricycle, bullet train, passenger car, pickup** |
|  | Devices 2 | **lawn mower, tractor, streetcar, tank** |

