# OpenReview forum: "Mid-Vision Feedback"
_ICLR.cc/2023/Conference — ICLR 2023 poster_

### Official Review · Reviewer_ym4H · 2022-10-24

**Confidence:** 4
**Correctness:** 3
**Technical Novelty And Significance:** 4
**Empirical Novelty And Significance:** 4
**Recommendation:** 8

**Clarity, Quality, Novelty And Reproducibility:**

The presentation format is rather strange, with a dump of figures and tables without any clear text accompanying them. Most of the tables and figures are not even cited, except for single lines in a 4-sentence discussion. Most of the paper is devoted to explaining MVF and basically a single page is devoted to the results, which is mostly a dump of tables and figures without any description or with minimal description.

**Strength And Weaknesses:**

Strengths

The proposed architecture is quite simple and therefore elegant.

The proposed ideas about how to incorporate feedback are rather generic and can be directly applied not just for any one specific dataset or task, but rather across multiple different problems including cases of no contextual cues, cases where contextual cues are incongruent or unusual, cases where contextual cues are provided by task demands or by statistical correlations previously learned from images.

Weaknesses

Several of the points below are questions, rather than weaknesses.

It is true that the majority of connections are feedback rather than feedforward, but not because of the work of Kveraga et al. In humans, we do not know any details about anatomical connectivity. Probably the best work making this point is Markov et al Cerebral Cortex 2014.

I find Figure 1 to be somewhat confusing. First, the authors state that they cropped the images to exclude context. But then they end up arguing that understanding the difference of context can help appreciate the low-level feature differences. Which one is it, is there context in these images or not? And does it help or not?

Figure 2. What are the injection sites? What are the big blue and red “A” symbols, are those the affine transformations? Are L_O and L_0 the same thing? Where do the high-level contextual expectations come from?

What are the x-axis and y-axis in Figure 5?

While this is not the main point of this study, why is it that ViT generally performs below VGG in Tables 3 and 4?
The rationale for having two stages in training is not clearly explained. Why two stages as opposed to a single stage of training?

Related to the question above regarding Figure 2, what decides which representations should be orthogonalized? In the example of desk and horse, who in the model says that there should correspond to two different contexts and should be orthogonalized?


**Summary Of The Paper:**

This is an interesting study that proposes novel ideas on how to incorporate contextual cues in the form of category-specific feedback signals into visual recognition algorithms. The proposed algorithm (mid-vision feedback, MVF) outperforms several baseline models in several relevant benchmark object recognition tasks. More generally, this paper introduces plausible mechanisms by which feedback signals (here in the form of expectations) could be incorporated into the traditional architectures for processing visual information.



**Summary Of The Review:**

This is an exciting effort to bring ideas about how feedback signals can be incorporated to help incorporate contextual cues into visual recognition. Although the clarity of the presentation can certainly be improved quite a lot, the results and ideas are novel and help push the field forward.

---

> ### Author Response · Authors · 2022-11-14
> **Reviewer Response**
>
> Thank you for your review! You are correct in your observation that MVF is generic, and therefor applicable to a variety of problems, datasets, and tasks. We anticipate that MVF can be taken in many directions in future work.
>
> Thank you for the reference to Markov et al.! We replace the previous reference with Markov et al. in support of top-down connections in biology.
>
> Regarding Figure 1, you are correct that the images are cropped in a such a way (closely cropped on the objects) that context is not immediately apparent. The intent of the figure is to demonstrate the importance of context in object classification, as the cropped images are more difficult to recognize due to the removal of background context. Once we understand the context of the images (even though they are cropped to make context less apparent), the objects in the images become clearer. This gives intuitive motivation for the importance of context in visual understanding. To rephrase: the images are cropped to minimize context, but bringing an understanding of context to the images makes them clearer.
>
> We adopt a lightweight variant of ViT (MobileViT) as referenced in the citation. VGG performs slightly better than MobileViT over CIFAR10 according to \href{https://github.com/pprp/pytorch-cifar-model-zoo}{https://github.com/pprp/pytorch-cifar-model-zoo} (93.54\% vs 92.9\%), in accordance with our runs over CIFAR100. We add a sentence clarifying this.
>
> "The rationale for having two stages in training is not clearly explained. Why two stages as opposed to a single stage of training?" - good question! We find that the network performs better when feedback is introduced after the network has had a chance to converge. Feedback introduces greater dynamic complexity into the training of the network, and introducing this complexity too early in training appears to hamper early learning.
>
> Injection sites correspond to the point in the network in which affine transformations are applied. You are correct in that the "A" symbols within the parallelograms in Figure 2 correspond to the affines we apply. We clarify this within the figure's caption. The figure and caption have the same orthogonal loss symbol $L_O$ (and $L_0$ was a typo - thanks for catching that! - it should be $L_O$).
>
> "Where do the high-level contextual expectations come from?" - good question. Conceptually these expectations could come from many sources. In our experiments we present performance w.r.t. two sources of context expectations: 1) ground truth, 2) predicted context. Experiments with ground truth context provide an assessment of how well MVF performs with perfect context knowledge. Predicted contexts are first generated by the same network MVF is running on, before being used via the MFV mechanism in a second run of the network to produce object category predictions. This provides information on how MFV performs in more realistic scenarios where the context needs to be predicted.
>
> "Related to the question above regarding Figure 2, what decides which representations should be orthogonalized? In the example of desk and horse, who in the model says that there should correspond to two different contexts and should be orthogonalized?" - good question. CIFAR-100, for example, provides super-classes which group object classes (we also apply these super-classes over imagenet), and CUB-200 provides an attribute ontology for birds. We use these ontologies in defining the contexts over which MVF is trained.
>
> Thank you for bringing the missing axis labels for Figure 5 to attention. The x and y axis of Figure 5 correspond to training iterations and loss value, respectively. We include these details within an updated figure.
>
> We expand discussion of results to improve clarity.

---

> ### Author Response · Authors · 2022-11-19
> **Updated Paper**
>
> Thank you again for your suggestions for improving the paper! Just an FYI, the updated draft with your suggestions is posted.

---

### Official Review · Reviewer_zASa · 2022-10-24

**Confidence:** 5
**Correctness:** 3
**Technical Novelty And Significance:** 2
**Empirical Novelty And Significance:** 2
**Recommendation:** 5

**Clarity, Quality, Novelty And Reproducibility:**

The work is a bit difficult to understand with regards to flow. I understand where the authors are going, but am not entirely convinced I got there in a clear way, so I perhaps could have missed something in my review that could change my mind. I'm also not sure why authors did not select neuroscience as the primary category for review and instead opted for "Applications"

**Strength And Weaknesses:**

See below for Main Paper Summary. TLDR: this paper studies mid-level vision as an improvement gateway for modern CNNs, however I do not think the baselines or experiments are well controlled enough to validate the statement. In addition it is not obvious what the contributions to performance are that stem from noise injections, recurrence and SSL-like learning when isolated, and not stacked together. In addition authors only focus on performance and not other (perhaps more interesting) tasks that extend object recognition such as common corruption robustness or adversarial robustness.

**Summary Of The Paper:**

This paper provides a mid level vision feedback module as an Add-On for CNN's. Authors argue that such mid-level feedback properties improve CNN performance (though perhaps for the wrong reasons), and also argue that this idea is worth pursuing given neuroscience/perceptual psychology motivated ideas.

**Summary Of The Review:**

++ Additional Controls and tentative confounding variables:

Tables 2 and 3 at first glance seem quite exciting and one could reasonably say that feedback is playing a critical role here. However, how can we dissociate between the effects of feedback and the effects of SSL or noise injection in training when comparing to the "vanilla" VGG, CNN and Transforer models? If a feedback loop would be the only other mechanism added in training then these tables are good indicators for the feedback effects in the paper, but if not, then knowing the exact contribution to the differential offset is not clear. Consider the question: what would have happened if a regular feedforward (withour recurrence) CNN would have been trained with noise injections, and PCA? Would the authors still get those positive bumps in score?

++ Several Missing References that hurt the paper in relation to where it lies on innovation:

Mid-level vision:
* Long, Yu & Konkle : "Mid-level visual features underlie the high-level categorical organization of the ventral stream" (texforms). PNAS 2019.
* Jagadeesh & Gardner. "Texture-like representation of objects in human visual cortex" . PNAS 2022.
* Harrington & Deza. "Finding Biological Plausibility for Adversarially Robust Features via Metameric Tasks". ICLR 2022.

Recurrence:
* Tang et al.: "Recurrent computations for visual pattern completion". PNAS 2018.
* Lotter, Kreiman & Cox: "Pred-Net". ICLR 2017.

Contrastive Learning:
* Konkle & Alvarez. "Instance-level contrastive learning yields human brain-like representation without category-supervision". ArXiv 2020 & Nature Communications 2022.
* Li et al. "Contrastive Clustering". AAAI 2021.

++ Additional experiments:

In experiments suggested by Long et al, Jagadeesh & Gardner, & Harrington & Deza, authors played around with machine vision classifiers showing them "texforms", these stimuli are stimuli that were synthesized from noise that match the local texture statistics of an object, in addition to preserving it's coarse shape. I wonder, if the argument here is to be made that mid-level feedback will improve general consistency and image categorization removing higher-order contextual biases, then the MVF-based networks should show similar confusion matrices for such texform stimuli and also the original stimuli.

While it may be too difficult for authors to add these experiments in the rebuttal, it would be nice to see these or some variation such as common corruption robustness (ImageNet-C) or adversarial robustness (either quantiatively or qualitatively : see for example Berrios & Deza, ArXiv 2022 on testing 'brain-alignment' of transformer based models).

+ Final Thoughts:

Overall, I don't think having a small one or 2 point percentage increase in classificatiion accuracy (albeit a very coarse control, unless I missed something) is enough to convince me that "feedback is better/necessary for computer vision". I am however on the author's side, and I believe their premise is true, but I don't think these experiments drive the point home completly.

---

> ### Author Response · Authors · 2022-11-14
> **Reviewer Response**
>
> Thank you for the additional references for related work! We incorporate these references into the paper. You're right that there is a strong connection between this work and its neuroscience motivation.
>
> With consideration to improvement margins when feedback is included, tuning experiments we've run since the paper deadline show greater margins.
>
> Your suggestion of exploring the relation between MVF and different input corruption is a compelling one, and one worth exploring. We expect that you are correct that MVF feedback can help with removing higher order noise in the input, and help counter the injection of bias in the input, if it is known what context such bias is associated with. Though a full consideration of this direction is beyond the scope of the present paper, we are endeavoring to provide performance numbers for Imagenet-C as you propose.
>
> Your suggestion of disentangling factors related to MVF is valid. We propose the following experiments to evaluate elements of MVF:
>
> SSL: We can report numbers when the context training signal is included in training, but when the feedback mechanism is not incorporated (neither the affines nor the orthogonalization loss). This will isolate the impact of the inclusion of the context label during training. The results will then be compared to models which include neither the feedback mechanism nor the context training signal, and the difference in performance will be attributable to the inclusion of the context signal during training.
>
> Noise injection: To evaluate the effect on performance due simply to the introduction of the feedback mechanism (and the random noise introduced by its introduction), but not due to the context training labels, we will report the numbers from experiments where the affine operations are included in the network but not trained.

---

> ### Author Response · Authors · 2022-11-19
> **Updated Paper**
>
> Thank you again for your suggestions for improving the paper! Just an FYI, the updated draft with your suggestions is posted. FYI, Section E of the Appendix contains evaluation over ImageNet-C, and Section J of the Appendix includes ablation evaluating the impact of noise from the affine transformations. FYI, Section E of the Appendix contains evaluation over ImageNet-C, and Section J of the Appendix includes ablation evaluating the impact of noise from the affine transformations. Additionally, experiments trained for maximizing feedback benefit show large margins (up to 9.4%), and are reported in Section 4.3.

---

> ### Author Response · Authors · 2022-12-07
> **Stage 2 Discussion**
>
> Hi – just checking in: friendly FYI that the end of the stage 2 discussion period is December 12. Thank you again for your feedback! As you recommended, experiments have been run to evaluate the impact of noise injection, robustness to corruption over the input (imagenet-C), as well as the impact of the context training signal (SSL).
>
> The results over imagenet-C and the results evaluating noise injection are in the updated draft, and an evaluation of the impact of the inclusion of the context signal during training can be found here: https://drive.google.com/file/d/1jude0GrrSZXPAECyXrjo0sQYnkjbvMpM/view?usp=share_link
>
> Thanks again, and let us know if you have any questions! We appreciate the effort you are putting into helping us improve this paper as a reviewer.

---

> > ### Comment · Reviewer_zASa · 2022-12-08
> > **Updating Score, thanks for clarification**
> >
> > Dear Authors,
> >
> > Thanks for providing additional updates. I believe the additional citations make the paper more complete and so do the extra experiments in the supplement. However I am still struggling similar to Reviewer ZiEu on understanding the more precise connections of the paper with mid-level vision. All-in-all, I am quite on the fence with this paper -- one on end, it would be great to see papers talk about these ideas at ICLR, but on the other, I am also looking at a paper that implements mid-level vision very vaguely.

---

> > > ### Author Response · Authors · 2022-12-09
> > > **Stage 2 Discussion**
> > >
> > > Thank you for your response! We're glad to hear that you believe that the feedback idea would add value to discussion at ICLR, and that the paper is now more complete with the additional references and experiments.
> > >
> > > We agree with you that there were several points of vagueness raised by reviewer ZiEu in the original draft.
> > >
> > > We clarify all implementation details in our response to reviewer ZiEu as well as the updated draft. Reviewer ZiEu had questions regarding several implementation-level details, which we clarify in our response to reviewer ZiEu as well as the updated draft. Reviewer ZiEu inquired about the application of affine transformations - whether "there [are] two matrices for the affine transformation, one for one context and one for the other" - we clarify that this is the case. Reviewer ZiEu also inquires about parameter counts - we clarify that there is a small bump in parameters when affine transformations are introduced. Reviewer ZiEu further asked where the context splits came from - we clarify that splits are derived from the existing label ontologies of the datasets used.
> > >
> > > Please let us know if there are further points of vagueness that can be clarified.

---

### Official Review · Reviewer_ZiEu · 2022-10-27

**Confidence:** 4
**Correctness:** 3
**Technical Novelty And Significance:** 3
**Empirical Novelty And Significance:** 3
**Recommendation:** 6

**Clarity, Quality, Novelty And Reproducibility:**

Clarity: The writing could be improved considerably. I provide a lot of typos and wording suggestions below, as well as a few slightly more substantive comments. Since you never talk about the contexts in the main paper, it would help a lot if you at least focused on one context split and explained what the context split was, and described how it helped classification. Figure 6 is a good start, but it would help to explain what the context is here. I see you say the first five and the last five are the two contexts, but that context split doesn't seem particularly well-motivated. Why is rocket with streetcar? Also, the "Absent" row and column is not explained, and it is apparently all 0's anyway, so I would remove it from the plot.

Quality: The experiments seem competently done.

Novelty: as noted above, I don't know of any other model that uses top-down feedback in this way; I think it is pretty novel and effective.

For reproducibility, I assume if this is accepted, the authors will provide their code, although this is not explicitly promised in the paper text. There are occasional contradictory statements about parameters and training set splits in the paper, which reduces reproducibility.

Wording suggestions, typos, minor comments:

In the abstract, you make the claim that "applications [of this idea] range from image and video understanding to explainable AI and robotics." You demonstrate that it is applicable to images, but the rest seems like a stretch without more justification. I would remove this and if you have room, put some discussion of how it would apply to these domains in the discussion section (4.4).

p2: "feedback naturally allows for detection of out-of-context objects" I don't follow this. It seems like contextual feedback would prevent the recognition of objects that don't fit the context. Please explain your reasoning here.

This opens up a couple possibilities -> This opens up a couple of possibilities

p2, reference to Figure 7: It would be helpful to have some form of Figure 7, e.g., a smaller one, right here to illustrate the point, as readers may not search ahead for Figure 7, and it isn't very clear until one gets to Figure 3.

Also in this paragraph, don't start a sentence with a lower case abbreviation ("w.r.t.") Instead write "Also, w.r.t. point #2,..."

In this page and the next, you italicize "injection level" three times. Once is enough, as long as you define it the first time, which you do.

"The fact that these vectors have been trained with a bias towards orthogonality w.r.t. contexts means that, through affine transformations, characteristics associated with context presence or absence can be manipulated with less collateral impact on other features."

reword as->

If these vectors have been trained with a bias towards orthogonality w.r.t. contexts allows for affine transformations to manipulate features associated with context presence or absence with less impact on other features.

"The fact that" suggests we already know this fact. Here, you are trying to say what the implication of this training is.

"Orthogonalization bias introduces a tendency towards orthogonality..." Of course it does, by definition! Say *why" you are trying to orthogonalize. In particular, you want to say something like, "The orthogonalization bias is introduced to increase the independence between contexts, so they can be manipulated without interference between them. [This point would be enhanced by having a version of Figure 7 here which shows the effect of this bias, making correlated context vectors (animate/inanimate, for example) more orthogonal.] You do say this, but this version makes the point immediately.

"This approach to incorporation of context expectations is controlled." Not sure what this sentence is saying.

Section 2.2: "course" -> "coarse", and "Additionallym" -> "Additionally"

The paragraph just before the methods section reviews irrelevant literature. This isn't the kind of feedback you are talking about, so eliminate this paragraph. This will free up some space to add something like Figure 7 earlier.
Caption of Figure 3: "and the injection level feature representations for those contexts produced by the injection level." ->
"and the injection level feature representations for those contexts."
"After feature vector modulation through application of an affine," ->
"After feature vector modulation through application of an affine transformation" (also in other places. "affine" is an adjective, but you keep using it as a noun.)

p6 of CIFAR100 or -> of CIFAR100 are

Tables 2,3,4: all tables: remove the percent signs and say they are all percents in the caption. Table 3, replace Pred Feedback with PF and explain in the caption; Table 4: Replace GT Masking and GT Feedback with GTN and GTF and describe in the caption; all tables: round to the 1st digit past the decimal, and you should be able to fit these in the width of the text.

lambda in one place is changed to ILS in Figure 5. Also, in the caption to Figure 5, you say you use 0.075 in all of your experiments, but in the 3rd line of section 4.1 you say you use 1.0.

p8, section 4.1.3: third line: "third last" attention block?

Appendix section B.1: "For full class breakdown see Appendix." Since you're already in the appendix, give the section in the appendix.

Section B.2:  "we designate 80% of the images in each class for training, but only *2%* for testing due to the computational cost incurred by the high number of images and the need for frequent testing."
Section F.1: "Like the Imagenet dataset, we designate 80% of the dataset for training and *20%* for testing."
Note the difference.

Also Section F.1: "Splits based on migration behavior, splits based on trophic level () and splits based on primary lifestyle."

Figure 7: Is this an artists conception, or based on data?

Figure 8: It would be a bit more helpful to notate the injection level in the figure.

"Splits based on migration behavior, splits based on trophic level () and splits based on primary lifestyle." Can you please be more specific?


**Strength And Weaknesses:**

Strengths:

+ This is a relatively novel idea about how to use top-down connections to improve performance in particular contexts. I have not seen any other work like this.

+ The context manipulation almost always improves classification performance.

+ It seems the context can be discovered automatically by the same network that is performing the classification, but the system usually works better if the ground truth context is known, which could come from some other process providing contextual feedback.

+ The approach is applied to a cross product of three different datasets and three different network architectures (except one cell is missing for a reasonable reason...), and almost always improves performance.

+ The approach is shown to work better than post-hoc filtering of the non-contextual outputs, at least when the ground truth context is known, but not when the context is predicted automatically.

Weaknesses, with concrete, actionable feedback

- The writing could be clearer

- It isn't clear whether the number of parameters is matched between the networks with feedback (which adds the parameters of the affine transformation) and without feedback. I assume this would add n^2 parameters, if there are n units in the layer.

- The context splits seem particularly arbitrary (e.g., streetcar and rocket are in one context?). It would be good use (for example) Figure 1 as a running example, and have a context that has food and animals or something similar and show that those images don't get confused in context.

**Summary Of The Paper:**

This paper proposes that images will be better processed if the context is known. Figure 1 provides excellent motivation for this idea. The images are very similar in appearance, but correspond to very different classes (food vs. animals). Inspired by the well-known fact that there are as many or more feedback connections in the visual system as feedforward ones, the paper proposes the idea that this feedback from higher levels can provide context information, altering the features to be more aligned with that context. So the idea of the paper is to provide biasing feedback to the mid-level features of the network, emphasizing one context (stretching a vector representing that context, while shrinking another). In order to avoid too much interference between feature dimensions when this transformation is applied, the network first is trained with an orthogonality bias at the particular layer where the context transformation will be applied (termed the "injection layer").  Then it is trained to use a biasing context, which is actually a learned affine transformation of the features at that layer of the network, controlled top-down. Hence, if the context is known, this can be applied dynamically, depending on the context. This contextual "hint" improves classification performance of the network. They note that this context manipulation could come from a more symbolic level of processing, providing a link between convnets and neuro-symbolic processing, although that itself is not demonstrated. They also show that post-hoc filtering of the results (masking off animal outputs in a food context, for example) doesn't work as well as altering the feature dimensions at a hidden layer. They show that this manipulation is relatively generic, and can be applied to a vision transformer as well as a convnet, improving performance in both cases.

I have read the authors' responses to my review, the other reviews, and skimmed the revised paper, and have bumped up my score by one point as a result.

**Summary Of The Review:**

This paper suggests a novel method for providing contextual feedback to a network that is performing categorization, inspired by the biology that there are many feedback connections in cortex. So the motivation is good. The results are that performance is almost always improved by context feedback. However, the actual implementation of this is a bit unclear. Are there two matrices for the affine transformation, one for one context and one for the other? I don't recall seeing this in the paper anywhere. It also isn't clear whether there is a matched number of parameters between the networks with and without contextual feedback. For these reasons, my enthusiasm for the paper is somewhat reduced.

---

> ### Author Response · Authors · 2022-11-14
> **Reviewer Response**
>
> Thank you for your detailed corrections and suggestions for the text! These are very helpful, and we are incorporating your suggestions, and restructuring. These suggestions are substantive and will considerably improve the quality of this draft. And yes, you are correct - code will be released upon acceptance. We hope you will find the newer version of the paper to be much improved over the original submission.
>
> To answer a few of your questions:
>
> You are correct that the affine transformations are NxN matrices (plus a bias), where N is the channel count of the injection layer. Proportionally this is a small increase in parameters over the models where feedback is not incorporated, as the affine transformations have 11\%, 6.25\% and 11\% (for VGG, SimpleCNN, and ViT) of the parameters of the layer immediately prior to injection. This amounts to the affines adding 1.4\%, 1.16\%, and 4.16\% (for VGG, SimpleCNN and ViT) of the parameters of the entire base network.
>
> We apologize for the confusion regarding the ILS value - the value of $0.075$ was chosen in a previous iteration of our experiments, and has been removed since.
>
> Good point about the Related Work - we remove the paragraph in question. This does free some space, and with space constraints in consideration we endeavor to fit a variant of Figure 7 into the Introduction.
>
> "Figure 7: Is this an artists conception, or based on data?" - this is an illustration to convey the intuition of decoupled representations, and not based on data. A more in depth consideration of decoupled representations can be found here: \href{https://arxiv.org/pdf/1804.08071.pdf}{https://arxiv.org/pdf/1804.08071.pdf}
>
> "Figure 8: It would be a bit more helpful to notate the injection level in the figure." - good point!
>
> You have a good point about the context splits - most of the context splits were taken from the superclasses of CIFAR-100 ( \href{https://www.cs.toronto.edu/~kriz/cifar.html}{https://www.cs.toronto.edu/$\sim$kriz/cifar.html} ), which we select as this is a standard benchmark ontology. We also include results for an \textbf{animate} / \textbf{inanimate} context split, which is a division we introduce - we update the manuscript to use this context split as a running example for clarity.
>
> And, good question regarding Figure 6! This context division comes from the super-class ontology of CIFAR-100. We select this split to show in the confusion matrix comparison, rather than the animate / inanimate split, as it has only 10 classes, while the animate / inanimate split has 100 classes and is too large to display in a figure. Good point about the "Absent" row and column - we will remove these from the figure.
>
> Your question of whether there is one or two affine matrices is valid, and we make it more clear in the captions of Figures 2 and 3 and in the body of our Methods section. There is one affine for each context, so two affines in settings with two context divisions and three affines in settings with three context divisions.
>
> With the phrasing "feedback naturally allows for detection of out-of-context objects" we were trying to convey that feedback, while it biases detection according to expectations, still allows for cross-context predictions, which contrasts with hard methods such as masking out detections from unexpected contexts, which do not allow cross context detections. Methods which do not exclude the possibility of cross context detections are useful, as on occasion objects occur in unusual contexts.
>
> ""Splits based on migration behavior, splits based on trophic level () and splits based on primary lifestyle." Can you please be more specific?" - good point. We provide a more detailed specification of these splits.
>
> Thank you for your suggestions on improving the readability of Tables 2, 3, and 4 - yes, this makes the tables clearer!

---

> ### Author Response · Authors · 2022-11-19
> **Updated Paper**
>
> Thank you again for your suggestions for improving the paper! Just an FYI, the updated draft with your suggestions is posted.

---

> ### Author Response · Authors · 2022-12-06
> **Stage 2 Discussion**
>
> Hi – just checking in: friendly FYI that the end of the stage 2 discussion period is December 12. Thank you again for your detailed feedback! The draft has been updated accordingly, and is much improved as a consequence.

---

> ### Author Response · Authors · 2022-12-09
> **End of Discussion Period**
>
> Hi - just checking in again with a friendly reminder that the final discussion stage ends December 12th (Monday). Thank you!!

---

> ### Author Response · Authors · 2022-12-10
> **End of Discussion Period**
>
> Dear Reviewer ZiEu,
> Please check our reply. Thanks!

---

### Decision · Program_Chairs · 2023-01-20

**Decision:**

Accept: poster

**Justification For Why Not Higher Score:**

The idea is novel and of wide interest, the resulting method works, but it could use more refinement for a higher score. The results, as reviewers point out, are an improvement, but not a major leap forward.

**Justification For Why Not Lower Score:**

This is clearly a novel idea: encode feedback by directly and systematically manipulating midlevel feature maps in a CNN. And it is convincingly demonstrated with experiments. That alone merits acceptance.

**Metareview: Summary, Strengths And Weaknesses:**

Summary: A new method to include top-down context while networks process images. Rather than adding feedback connections, this method manipulates the midlevel feature maps of the network directly.

Strengths: The reviewers had not seen a similar approach. This idea could be applied to other algorithms and could easily be built upon. The method also provides an additional level of interpretability where one can perform meaningful manipulations of the internal state of the CNN. Between the generic nature of the method and its novel idea, this is of clear interest to the community.

Weaknesses: Reviewers found the manuscript a bit difficult to follow at times. This is likely because of a protracted motivation section rather than focusing on the method immediately; the description of the method begins on page 3. An early crisp description of the method, that is factual and straightforward without context or embellishment, like that provided by reviewer ZiEu would improve the manuscript.


**Note From Pc:**

if the above contains the word "oral" or "spotlight" please see: "oral" presentation means -> notable-top-5% and "spotlight" means -> notable-top-25%. As stated in our emails, we are disassociating presentation type from AC recommendations

**Summary Of Ac-Reviewer Meeting:**

N/A